# The Correlation between Vehicle Vertical Dynamics and Deep Learning-Based Visual Target State Estimation: A Sensitivity Study

**DOI:** 10.3390/s19224870

**Published:** 2019-11-08

**Authors:** Yannik Weber, Stratis Kanarachos

**Affiliations:** Research Institute Future Transport and Cities, Coventry University, Priory Street, Coventry CV1 5FB, UK

**Keywords:** automated vehicles, object detection, object tracking, distance estimation, road anomalies, road bumps

## Abstract

Automated vehicles will provide greater transport convenience and interconnectivity, increase mobility options to young and elderly people, and reduce traffic congestion and emissions. However, the largest obstacle towards the deployment of automated vehicles on public roads is their safety evaluation and validation. Undeniably, the role of cameras and Artificial Intelligence-based (AI) vision is vital in the perception of the driving environment and road safety. Although a significant number of studies on the detection and tracking of vehicles have been conducted, none of them focused on the role of vertical vehicle dynamics. For the first time, this paper analyzes and discusses the influence of road anomalies and vehicle suspension on the performance of detecting and tracking driving objects. To this end, we conducted an extensive road field study and validated a computational tool for performing the assessment using simulations. A parametric study revealed the cases where AI-based vision underperforms and may significantly degrade the safety performance of AVs.

## 1. Introduction

The functionality of automated driving systems (ADS) is grounded on a processing chain of perception, control and vehicle platform manipulation [1]. For data gathering, camera sensors pose the most cost-effective solution and thus have found their way into most ADS [2]. Camera-based vehicle perception can be divided into three levels of complexity that build on each other: Object detection, tracking and behavior analysis [3,4]. Object detection is in turn divided into appearance- and motion-based solutions [5]. While motion-based approaches rely on a subject’s motion signature in a continuous image stream [6,7], appearance-based detectors are again subdivided into one- and two-step procedures [8]. All stages are illustrated in Figure 1. One-step detectors such as You Only Look Once (YOLO) gained popularity due to less complex processing pipelines, high frame rates and accuracy [9,10]. Two-step algorithms on the other hand perform hypothesis generation (HG) and verification (HV) to increase detection reliability [11]. For HG, in a first step local feature descriptors such as Histogram of Oriented Gradients [12], Haar-like features [13], shadow-based intensity gradients and thresholding [14], deformable part models [15] or Aggregate Channel Features [16] are used to specify the region of interest (ROI). HV approaches for hypothesis confirmation include boosting algorithms such as AdaBoost [17,18], Support Vector Machines [19,20], or deep learning systems such as Convolutional Neural Networks (CNNs) [21], very deep CNNs [22] or region-based CNNs (R-CNN) such as Faster R-CNN [23,24] or Mask R-CNN [25]. Apart from typical two-step detectors, other techniques use e.g., pavement or lane detection for ROI alignment followed by illumination variation or edge filters to verify potential object candidates [26,27], a two-stage regression procedure [28] or the combination of appearance-based HG and motion-based HV with Haar-like features and AdaBoost [29].

In a second step, object trackers are applied to understand the detected object’s inter-frame dynamics and motion characteristics. This can be achieved by extracting vehicle shape information and creating an inter-frame motion vector [30], using particle filtering [31], nonlinear observers [32], color probability density calculation [33] or Kalman Filters (KF) [34]. Further surveys have analyzed and summarized a variety of procedures [35,36].

Similar to the human driving process and in addition to information on surrounding objects’ size, shape, speed and relative position, a vehicle must accurately perceive the distance from different road objects to generate safe driving envelopes [37]. For current ADS such as Forward Collision Warning or Automated Emergency Braking, the distance is essential to calculate the time to collision [38]. Several estimation methods have emerged over time including intrinsic camera parameters combined with target width estimation or ground-plane camera angle-based approaches. One proposed system makes use of a combination of vertical and horizontal triangulation to translate the real world into camera coordinates. Here, the size and distance to the detected object is calculated with known internal camera parameters such as focal length and focal point position [39]. However, vehicle width-based estimation techniques only achieve about 30% detection accuracy [40]. This issue can be compensated by adding a priori knowledge of the smallest and largest measurable distance between camera and target object [41]. Another solution to increase reliability is to use the predicted bounding box bottom-line center point and the vehicle’s pitch angle for triangulation [42]. To cope with varying pitch angles a virtual horizon can additionally be computed and used as a reference line [43]. Another method is measuring the top or bottom light ray hitting the camera’s imaging sensor to compensate sensor movement [44]. With a Lane Departure Warning system on board, the acquired lane information can be paired with vehicle classifiers to generate a solid ground-plane and design a more robust distance estimator [38,45]. Optical flow measurement combined with system controller input information can also serve the estimation procedure [46].

### State-of-the-Art

Although each method contributes to the realization of reliable monocular tracking and distance estimation systems, they share disadvantages in real-world scenarios. While some approaches introduce compensation measures for small pitch angle and surface envelope variations, a thorough search of the literature revealed a gap in compensating large and rapid vertical dynamics that influence ADS performance. In the domain of road surface irregularities, contributions stretch into the areas of online road surface estimation and automated image analysis for maintenance purposes. Within the first field, studies include the analysis of the vehicle’s *z*-acceleration to recognize potholes [47,48], ultrasonic road anomaly detection [49,50] or the observation of a preceding vehicle to estimate the road anomaly severity based on the target chassis movement or the activation of brake lights [51,52]. In the second area, research was conducted to identify rugged road sections with Machine Learning (ML) algorithms such as CNNs [53,54,55].

In the time consuming process of proving the safety of the intended functionality, coping with all the challenges is infeasible in real-life experiments, which is why the testing, validation and homologation of assistive systems is shifted to a virtual environment [56,57]. Here, it is possible to cover a variety of corner cases without increasing time and cost and the risk to cause harm to drivers, vehicles and the environment. In the virtual validation process, it is mandatory to calibrate the system specifications and the simulation environment with latest test results to constantly track the system performance and complexity. Gradually increasing the level of simulation realism and adding a feedback loop to the design and testing stage guarantees early detection of both architectural and simulation boundaries [58]. Both ISO 26262 and ISO 21448 provide general guidelines for X-in-the-Loop development, verification and validation stages [59,60]. Complexity increases with the application of ML algorithms in the vehicle’s perception process. As the systems solely rely on images, training data for the object detector must contain all system requirements. Even small input data deviations in between test cycles have the potential to deteriorate perceptual performance [61].

In this paper, we conduct a thorough and systematic computational study on the influence of road anomalies on the environmental perception pipeline of modern ADS, including the stages of object detection, tracking and motion and distance estimation. To this end, we generate two fully validated vehicle simulation and camera sensor models: A multi-body vehicle dynamics and monocular camera representation in the IPG CarMaker (CM) environment are generated by using results of controlled real-world experiments [62]. For vehicle detection we apply the state-of-the-art ML networks YOLOv3 and Mask RCNN [25,63]. Targets are tracked with a combination of the Global Nearest Neighbor (GNN) method and a KF to predict future object positions [34,64]. With an intrinsic camera pinhole model the distance between the ego vehicle and the target’s bounding box bottom-line center point location is computed [42]. To the best of our knowledge this is the first time a study on the impact of vehicle vertical dynamics on the performance of vehicle detection, tracking and distance estimation is performed. It is also the first contribution applying a perception pipeline based on state-of-the-art deep learning-based object detectors to a fully validated IPG CarMaker simulation environment to accelerate the verification and validation of automated driving functions in the system development process.

The rest of the paper structures as follows: After describing methods and tools used in our research, we present the experimental results from several real-world driving trials. Then, the computational results of our study are analyzed, followed by conclusions where the advantages of our proposed methodology are highlighted and finally the scope of upcoming projects and future research directions.

## 2. Materials and Methods

### 2.1. The Perception Process

#### 2.1.1. Object Detection

For the task of object detection, the two neural networks YOLOv3 and Mask RCNN were used. The algorithms were released throughout the previous year and achieve high mean average precision measures of 57.9 at 78 frames per second (fps) (YOLOv3) and 60 at 5 fps (Mask RCNN) [25,63]. While YOLOv3 only predicts the target’s bounding boxes, Mask RCNN performs semantic segmentation, allocating every pixel within the image to the object class. Both networks were trained on the Common Objects in Context (COCO) data base, consisting of 2.5 million labelled objects [65].

#### 2.1.2. Object Tracking

After recognizing an obstacle in the recorded video frame, a bounding box is annotated to determine the obstacle’s location in the image. Now, the Intersection over Union (IoU) of the detected box and a predicted box (the gate) is measured to then calculate a cost matrix containing all IoU values for each detection made [66]; the bigger the IoU value, the lower the cost. The cost represents the distance between the actual detected box and a predicted one. The aim is to minimize the cost and therefore the distance to make predictions as accurate as possible. Here, the GNN algorithm combines overlapping detections to unified clusters where every independent detection receives its own cluster. Secondly, the cluster with the lowest cost is assigned to a previous track to ensure a continuous tracking stream. Object identifiers (IDs) are now forwarded to the dedicated confirmed track. In case of a new cluster, a new track is initialized with an original ID. Finally, a discrete, 2D time-linear KF is used for position estimation and gate prediction for the next frame. In this fundamental approach, the velocity is assumed to be constant (zero acceleration) to show the effects non-steady states can have on the system performance and to derive new corner cases that must be taken into consideration for SOTIF. Furthermore, a frame rate of 30 fps allows assuming the velocity to be steady [67]. The system state $\hat{X}$ is declared in Equation (Equation 1) containing the *x*-position, *x*-velocity, *y*-position and *y*-velocity of the tracked vehicle, as well as the bounding box-width, the velocity of width changes, the height and the velocity of height changes. The measurement input $\hat{Y}$ is displayed in Equation (Equation 2) including the longitudinal and lateral position and the target proportions in height and width. The state transition matrix *A*, measurement model *H* and system error *Q* are presented in Equations (3) to (5).
(1)$$\hat{X}=\left[x\;v_{x}\;y\;v_{y}\;w\;v_{w}\;h\;v_{h}\right]$$
(2)$$\hat{Y}=\left[x\;y\;w\;h\right]$$
(3)$$A=\left|\begin{array}{c} 1\;dt\;0\;0\;0\;0\;0\;0\\ 0\;1\;0\;0\;0\;0\;0\;0\\ 0\;0\;1\;dt\;0\;0\;0\;0\\ 0\;0\;0\;1\;0\;0\;0\;0\\ 0\;0\;0\;0\;1\;dt\;0\;0\\ 0\;0\;0\;0\;0\;1\;0\;0\\ 0\;0\;0\;0\;0\;0\;1\;dt\\ 0\;0\;0\;0\;0\;0\;0\;1 \end{array}\right|$$
(4)$$H=\left|\begin{array}{c} 1\;0\;0\;0\;0\;0\;0\;0\\ 0\;0\;1\;0\;0\;0\;0\;0\\ 0\;0\;0\;0\;1\;0\;0\;0\\ 0\;0\;0\;0\;0\;0\;1\;0 \end{array}\right|$$
(5)$$Q=\left|\begin{array}{c} \frac{dt^{4}}{4}\;\frac{dt^{3}}{3}\;0\;0\;0\;0\;0\;0\\ \frac{dt^{3}}{3}\;dt^{2}\;0\;0\;0\;0\;0\;0\\ 0\;0\;\frac{dt^{4}}{4}\;\frac{dt^{3}}{3}\;0\;0\;0\;0\\ 0\;0\;\frac{dt^{3}}{3}\;dt^{2}\;0\;0\;0\;0\\ 0\;0\;0\;0\;\frac{dt^{4}}{4}\;\frac{dt^{3}}{3}\;0\;0\\ 0\;0\;0\;0\;\frac{dt^{3}}{3}\;dt^{2}\;0\;0\\ 0\;0\;0\;0\;0\;0\;\frac{dt^{4}}{4}\;\frac{dt^{3}}{3}\\ 0\;0\;0\;0\;0\;0\;\frac{dt^{3}}{3}\;dt^{2} \end{array}\right|$$

#### 2.1.3. Distance Estimation

The exact position of confirmed target tracks can now be determined. For proper application of the monocular camera used in this study, the intrinsic (camera hardware specifications) and extrinsic (camera position) parameters were extracted first. The parameters, in particular the focal length, focal point position in *x*- and *y*-camera coordinates and a distortion matrix (to model the distortion the camera lens layout causes in the acquired image) were obtained by calibrating the device with the MATLAB Single Camera Calibrator App [68]. All quantities mainly affected the representation of the acquired scene in the digital images. With these parameters, a projective transformation was performed where the location of an object in image pixels (bounding box) was transferred to real-world coordinates. The real width and height of a present object could then be defined as *X* and *Y* respectively. The corresponding values in pixels are represented by *u* and *v*. Finally, the distance could be computed together with the known focal length *f* and focal point $o_{x}$ and $o_{y}$.
(6)$$u=f*\frac{X}{Z}+o_{x}$$
(7)$$v=f*\frac{Y}{Z}+o_{y}$$

*Z* is the desired distance between the camera and the obstacle. In this case, the width of the object does not have to be known for the calculation as only the bottom-line center point of the extracted bounding box is used for the computation. It also serves as a reference to estimate the lateral translation *Y* of an object. Finally, the equations are re-organized to solve the variable *Z*.

#### 2.1.4. Real-World Driving Data

In a first step, we conducted controlled real-world driving trials to generate a comprehensive data set for the validation of vehicle dynamics and monocular vision. The most critical proprioceptive quantities here are the vehicle *z*-acceleration and the resulting pitch angle, responsible for the up- and downward chassis motion and eventually the distortion in the camera-recorded image data. A Ford Fiesta MK7 was therefore equipped with (1) a monocular full-HD camera recording images at 30 fps, (2) a Velodyne VLP-16 LiDAR scanner operating at 10 Hz for reference data generation, (3) a data logger to extract vehicle state information from the Controller Area Network (CAN) and (4) a smartphone mounted in the position of the camera to capture the *z*-acceleration the module is facing while riding over road bumps. The interested reader is referred to the data set publication for a detailed description of the data acquisition process, sensor data sheets and positioning. The full Coventry University Public Road Data Set for Automated Cars (CUPAC) - including data from a front-facing infrared camera and an additional smartphone camera - is accessible at [69], comprising of 105 GB of public road recordings in left-hand traffic around the city of Coventry in the UK. The data is split into 14 different scenarios, covering a variety of weather, road surface and traffic conditions. Furthermore, the set contains around more than 100 road bump objects with correlating tri-axial acceleration data and a multiple of the quantity covering anomalies with inertial vehicle measurements.

### 2.2. Vehicle Dynamics Model Validation

In the next part, two different IPG CM simulation models were validated separately to guarantee plausibility of the investigation and validity of the research results: (1) A multi-body vehicle dynamics and (2) a monocular camera vision model. In (1), the effects and influences of road anomalies on the computed vehicle dynamics response were to be isolated from longitudinal influences from the powertrain and braking system as well as lateral disturbances resulting from changing steering angles or road bumps approached at an angle smaller or larger than 0 degrees. Constraining the experiments was necessary to guarantee model validity and for an unfiltered view at the vertical vehicle dynamics effects which eventually had an influence on the vision module, leading to part (2) of the validation procedure, presented in Section 2.3.

#### 2.2.1. Dynamics Model Validation Criteria

This part presents the steps undertaken to validate model (1) using real-world measurements. A deductive top-down approach according to the recommendations of [70] was followed for a reliable validation procedure, presented in Figure 2. Highlighted in green, the validation criteria are first to be determined. The process is further divided into setting up the validity metrics and data handling procedure (yellow boxes) and finally the selection of maneuvers to imitate real-life driving conditions (red boxes), which will be explained in upcoming sections. For the definition of the validation criteria, defined pass/fail thresholds directly evaluate whether the current virtual parameter setup fits the real vehicle dynamics configuration to make a statement on model validity. The criteria highly depend on the specific use case, environment and aims of the study itself. The study was designed with reference to [71].

Simulation and real vehicle response variables recorded in different field trials were compared and evaluated for three independent output parameters each. In this study, the variables under consideration are the vehicle acceleration in *z*-direction $a_{z}$ and the pitch angular velocity about the vehicle’s *y*-axis $\omega_{y}$ affecting the rate the output camera images are changing depending on the time driven. In addition, the continuous ego-vehicle velocity $v_{abs}$ was observed to mitigate undesired longitudinal influences on the vertical dynamics [71]. For the identification of the simulation model parameters (vehicle and camera sensor models), linear regression models were fitted and formulated as a minimization problem using the method of least squares. Here, the aim was to find three functions that describe the correlations between the simulated (Y) and true recorded value (X) for each the *z*-acceleration, pitch angular velocity and vehicle absolute velocity. The fitness of the linear regression model was then computed by the coefficient of determination ($R^{2}$) and the root mean square error (RMSE), presented in Equations (8) and (9). To mitigate the negative influence of noise in the sensor output signals, the RMSE was employed as the objective function value to control value deviation and to ensure that valid statements on model accuracy could be made. As the focus of this study was on conducting a first generic investigation on the influence of vehicle dynamics on visual system performance, small deviations in the real response variables did not have a noteworthy influence on the plausibility of the results. The minimum pass-values for the system to achieve the status ’valid’ are summarized in Section 2.2.4.
(8)$$R^{2}=\frac{\sum {\left(Y_{i}-\bar{Y}\right)}^{2}}{\sum {\left(X_{i}-\bar{Y}\right)}^{2}}$$
(9)$$RMSE=\sqrt{\frac{1}{n}\sum_{i=1}^{n}\left(\right|X_{i}-Y_{i}{\left|\right)}^{2}}$$

A consistent threshold of 67% for $R^{2}$ ($R_{min}^{2}$) was chosen to guarantee at least two thirds of the real model were represented by the corresponding simulation output. The maximum RMSEs ($RMSE_{max}$) were set to 1 m/s^2^ for $a_{z}$, 0.1 s^−1^ for $\omega_{y}$ and 1 m/s for the velocity.

#### 2.2.2. Dynamics Model Validation Metrics

The next step following Figure 2 was setting up the simulation input parameters, or validation metrics, which determined the computed outcome discussed in the previous section. The metrics included all user-accessible quantities taken into consideration within each simulation iteration. With a focus on system performance on road bump obstacles, the vehicle mass distribution and suspension configuration were the defining quantities in the multi-body vehicle dynamics model. In particular, the research vehicle body mass *m* and moment of inertia $I_{\psi }$, the center of gravity distance to the front $l_{f}$ and rear axle $l_{r}$, the tire diameter $r_{e}$ and moment of inertia $I_{\omega }$ and the front $t_{\omega f}$ and rear track width $t_{\omega r}$ defined the already known static vehicle setup, acquired from a previous experimental study and presented in Table 1. The residual unknown parameters were the spring and damper characteristic curves. The standard IPG CM spring compression and damping velocity curves for a small hatchback similar to the research vehicle were adapted. The remaining tuning factors were the spring $k_{F/R}$ and damping constants $c_{F/R}$ on the front and rear axle determining the variable suspension setup.

For the variation of the input parameters a Design of Experiment (DoE) approach was followed as we had no a priori knowledge of the four independent suspension variables. The aim was to investigate all parameter variations to find a setup that accurately represented the real-world vehicle suspension and dynamical behavior. For better controllability and to limit the search space, the maximum deviation for each constant was amplified to $k_{F/R}=\left[0.5,2.1\right]$ and $c_{F/R}=\left[0.5,2.1\right]$ times the original setup.

#### 2.2.3. Ground Truth Scenario Generation—Road Anomalies

The last step for identifying the model parameters and obtaining a validated simulation model was designing the validation scenarios. Therefore, 5 scenarios only involving road bumps were identified by visually and computationally analyzing the recorded field trial data. On the one hand, the recorded mono camera images were scanned manually for road anomalies. On the other hand, the mean absolute deviation (MAD) of the *z*-acceleration streams extracted from the smartphone sensor sequences were calculated to detect outlying values which indicated road bump objects as well. As the vehicle’s Inertial Measurement Unit was not capable of recording the body frames’ *z*-acceleration, the smartphone’s tri-axial accelerometer measurements were used. Road bumps were identified when the *z*-acceleration exceeded three times the sequential MAD. In our test series, a MAD of 0.47 (m/s${}^{2}$) was reached. Thus, data points of acceleration values smaller than −1.41 (m/s${}^{2}$) and larger than 1.41 (m/s${}^{2}$) were extracted. A full scenario parameterization is presented in Table 2. In summary, the final test scenarios covered speed (*v*) ranges between 19 and 31 [km/h] with resulting *z*-acceleration ($a_{z}$) values from −3.5 to 3 m/s^2^ and pitch angular velocities (ω) from −0.5 up to 0.5 s^−1^. The road bump height varied between 0.08 and 0.12 m with an average length of 1 m.

#### 2.2.4. Dynamics Validation Results

The scenarios were now executed with every possible parameter combination initialized by the DoE. The validation criteria were met throughout the whole scenario set, as presented in Table 3. An exemplary overlay of real and computed *z*-acceleration response is illustrated in Figure 3. All regression lines can be found in Appendix A.

### 2.3. Vehicle Deep Neural Network Vision Model Validation

For the vision model, a realistic recreation of collected real-world scenes was necessary to create a 1-to-1 replica of the camera behavior and resulting image output. As both the object detection and distance estimation algorithms were solely relying on pixel input data, it was again crucial to isolate the vision from other disturbances first and focus on analyzing functional disparities between realistic and simulated data sets. Afterwards, the validated vehicle dynamics model from previous section was used to then combine both computational model parts to prove the dynamics and vision model valid. This part presents the procedure to validate an IPG CM monocular camera vision model. Analogue to previous section, the deductive validation approach was now transferred to the virtual replication of the real-world camera used in the conducted field trials and went according to Figure 4.

#### 2.3.1. Vision Model Validation Criteria

The systems to be considered for the vision model validation were the following: (1) The object detection on the one hand and (2) the distance estimation algorithm on the other hand. While both procedures made use of the image data generated by a real or virtual camera, the performance was dependent on different aspects and features of the used data. Starting with system (1), in a first step the detectability of real and simulated target vehicles was to be validated to ensure the detector interpreted real recorded and computationally generated data in a similar way. Therefore, the algorithm’s precision and recall were measured on both scenario sets and compared. Furthermore, not only the recognition accuracy in the image data was important, but also the target’s scene representation in terms of size and format. In addition to the physical appearance of detected vehicles, their position within the data was essential for the performance of algorithm (2). To validate the distance estimator, especially the bottom-line center point location of the generated bounding box had a huge impact on the correctness of the output distance value. It was validated by employing the LiDAR measurement data which provide high accuracy distance and velocity information. The adapted workflow according to Figure 2 is presented in Figure 4, where the pass/fail evaluation steps are highlighted in green color as well.

The validation criteria consisted of:The detector precision and recall on real and virtual targets. Here, the numbers of true positives, false positives and false negatives were evaluated.The real and virtual bounding box size difference, where both height and width were analyzed separately.The difference between the real and virtual distance estimation performance. After the distance estimator precision was validated using the ground truth LiDAR information, both accuracy values were compared.

The camera sensor model validation was conducted considering flat road scenarios.. Real and simulated results were compared and examined for similarity in a representative study. In all criteria, the minimum ’pass’ requirement was set to a maximum allowed real/sim-result deviation of 90%. For criteria 1 and 2, the average values were calculated for randomly sampled image sequences. In Criterion 3, the distance between ego and target vehicle was tracked over the whole duration of the real and simulated sequence. Finally, the performance was determined by calculating the error between the mean deviation and absolute distance value. All performance indicators were then averaged over the total number of indicators.

#### 2.3.2. Vision Model Validation Metrics

To generate the required simulation data, a virtual camera was attached to the simulated twin vehicle which was then configured to deliver realistic vision data. From the original hardware calibration, the *x/y/z*-position $d_{x}$, $d_{y}$, $d_{z}$ and rotation $r_{x}$, $r_{y}$, $r_{z}$ and the field of view FoV were included to the static vision setup and are given in Table 4. While the Matlab camera calibration procedure delivered information on distortion grid matrices for modeling the lens curvature and device characteristics, the IPG CM virtual camera could only be configured by the scale *S* of the recorded image and a first, third and fifth grade radial distortion configuration *r*, $r^{3}$ and $r^{5}$.

Another DoE was set up to recreate the mono camera characteristics that could not be directly fed into the simulation environment. *S* was modeled in the interval [0.5, 1.5], *r* in [0.5, 1.5], both with a step width of 0.2; $r^{3}$ in [−0.25, 0.35] and $r^{5}$ [−0.25, 0.35] in steps of 0.15.

#### 2.3.3. Ground Truth Scenario Generation—Scene Representation

Five different scenarios of 10 s run time each were selected to analyze the computational situational awareness and to validate the vehicle vision simulation model. Scenario 1 to 4 included road bumps, while scenario 5 was a flat road and used as a reference. The sequence included stationary and moving targets, as well as a broad spectrum of velocities and steering angles. The detectors had to master challenges such as occlusion, changing object recording angles and continuously varying vehicle states. It was used for the first camera calibration to isolate the vision system from vertical acceleration influences before approaching the combination of vision and dynamics in the upcoming stages. As the velocity has an impact on the variation of distance to other target vehicles, the derived scenarios corresponded to different vehicle speeds between 12 and 34 km/h. A scenario parameterization is given in Table 5. While the ego vehicle followed a moving target in scenarios 1 and 2, scenarios 3 and 4 solely included stationary objects. In scenario 5, ego and target vehicle were approaching. In combination the scenarios covered a variety of every-day driving situations.

The real-world scenarios were then replicated in the simulation environment. With the help of the collected LiDAR data, a digital twin based on the validated vehicle dynamics and camera sensor models was created including accurate target vehicle positions, speed profiles, road dimensions and distances to surrounding road furniture and buildings. For the object detection and distance estimation, ground truth data of the vehicle position in the images was generated by assigning the object’s bounding boxes. With the aid of the target pixel locations, the algorithm accuracy was then evaluated with the performance metrics introduced in previous chapter.

#### 2.3.4. Vision Validation Results

In the analysis of validation Criterion 1, a study was conducted to estimate whether state-of-the art vision systems would have the ability to detect objects in the simulation environment. Since the algorithms were solely trained on real-world images, there is a chance that the algorithm’s sensitivity to virtual data varies significantly. First, the state-of-the-art networks YOLOv3 and Mask RCNN were chosen to be evaluated using reference scenario 5 to solely concentrate on detection and tracking performance. Once these networks were proven suitable for the task, it could be ensured with sufficient evidence that the detector’s response in the simulated environment would be correlating with the results on field trial data.
(10)$$Precision=\frac{TP}{TP+FP}$$
(11)$$Recall=\frac{TP}{TP+FN}$$

Precision and recall, presented in Equations (10) and (11), were chosen as the key performance indicators for each network and calculated for the whole driving sequence. The detection distance relevant for the performance evaluation was limited to 30 m. Objects placed further apart appeared too small for the detection. As can be seen in Table 6, both algorithms achieved sufficient performance values with both precision and recall being consistently above 90%.

Coming to the evaluation of criteria 2 and 3, an object detection and distance estimation pipeline was developed in Matlab which was fed with the derived bump scenarios from previous section. It consisted of a bounding box generator and the distance estimator which were used for the overall performance evaluation for answering the research questions. Instead of an actual detector, the ground truth bounding box information was directly fed into the evaluation loop to mitigate effects of potentially unstable detection outcomes and to solely concentrate on the vision model representation. The procedure is summarized in Figure 5 and will be explained in more detail within the next section. For efficiency reasons, the DoE was executed on scenario 5 and then fine-tuned on numbers 1 to 4. Furthermore, for the distance estimation, only the detection of the moving target vehicle was relevant as the detection capability itself was already confirmed in previous study. All regression lines are attached in Appendix B with the validation results summarized in following Table 7. All criteria were met, both without and with the vehicle dynamics influences included.

### 2.4. System Environmental Perception Analysis

In this final section, the performance of the vehicle vision model in a variety of real-world scenarios is presented. The aim was to clearly outline how and to what extend riding over a road bump has an impact on the reliability of object detection, tracking and distance estimation algorithms and when the system exceeds its performance boundaries. In a first step it was necessary to define the key performance indicators that determine a system’s reliability. Two parameters were seen as most critical: The ego-vehicle velocity and the height of a road bump. Higher velocities lead to a bigger impact on road anomalies and therefore a rise in *z*-acceleration and pitch angular velocity, which are directly translated to the camera module. This movement in turn can affect the tracking and distance estimation performance.

#### 2.4.1. System Environmental Perception Metrics

As already introduced in foregoing paragraph, the first performance measure was the system dropout time. It was defined as the time span in seconds in which the estimated absolute distance value deviated more than 50% from the ground truth distance to the tracked vehicle. This deviation threshold allowed the assessment of the potential threat for public road safety [56]. Additionally, the vehicle tracker performance was analyzed by logging the minimum IoU value of the tracked vehicle’s ground truth and computed bounding box. The figure was essential to verify whether the vehicle system gained a sufficient understanding of the surrounding driving scene in terms of hazard location and motion prediction. In case the estimated image location differs significantly from the real target, the system response would not match the required actions for threat avoidance and safe driving envelope generation. Furthermore, the fusion of several sensor and detection systems relies on sufficient plausibility and matching of the combined sensory data.

#### 2.4.2. System’s Environmental Perception Boundaries Exploration

To identify the system’s boundaries, a systematic computational study was conducted as every used model represented real-world conditions to a sufficient extend. As illustrated in Figure 5, images were loaded step by step from a generated virtual video sequence. Then, the ground truth bounding box information of a moving reference target vehicle was fed to the object detection stage which substituted the use of an actual detection system. This step was undertaken since the proper functionality of two independent object detection algorithms was already sufficiently validated in Section 2.3.4. Additionally, it allowed to isolate the stages of object tracking and distance estimation from other influences and to mitigate fault propagation. Now, the information was processed to the tracking stage where continuous detections were assigned to consistent vehicle tracks. An important step was the motion estimation procedure, which predicted plausible future target positions and had a direct influence on the assigning process of upcoming bounding boxes. As presented in Figure 5, the finally assigned boxes were fed back into the tracking stage. In this stage, the first performance break-out point was installed where the difference of the tracked target location was compared to the ground truth position. While this step might appear redundant due to the ground truth bounding box being fed into the whole pipeline twice, it had a valid reason of existence here. As outlined before, the vehicle tracker consequently calculated plausible vehicle positions. If a current detection might therefore vary from the predicted location in the acquired sensor data, the algorithm prioritizes plausible predictions over the detector’s generated box information. On the one hand, this behavior can compensate possible detection dropouts for a certain horizon to guarantee for a steady information stream. However, if the vehicle state changes rapidly, as is the case on road bumps, the algorithm behavior might deliver faulty results, which exactly emphasizes the scope of this study.

Finally, the distance from the assigned track was calculated by observing the bounding box bottom-line center point. A second break-out was designed to verify the accuracy with the exact ground truth distance from LiDAR measurements. Both break-out quantities were then forwarded to a last performance analysis stage, the results of which are presented in the next section of this work. For the variation of the vehicle velocity and road bump size, another DoE was configured with a static vehicle to target distance of 30 m which in turn was the maximum distance the detector could deliver sufficient detection accuracy. The ego vehicle velocity was varied from 10 to 60 km/h in steps of 10 km/h, outreaching the highest validation value of 32 km/h and stating a potential limitation of the research presented. However, the validation was conducted with high fidelity, covering a variety of driving scenarios, which is why validity for higher speed ranges could be assumed to investigate and even exceed the regular speed limits of urban driving. On the other hand, the obstacle size was set to the proportions occurring in the field trials. Values of 0.08, 0.1 and 0.12 m were chosen for the road bump height. In total, the configuration led to 18 different scenarios that are discussed in the next section. A third set of scenarios was designed to study the actual research question. Only the ego and target vehicle were placed on a straight road with a central road bump obstacle (see Figure 6). Any undesired influence on the vehicle dynamics and vision system could be excluded from the evaluation procedure with this approach. The ego and target vehicles were given identical speed profiles to maintain a constant distance of 30 m, so the maximum distance to guarantee plausible detection results.

## 3. Results

The results of the parametric analysis are presented in this section. Heat maps were chosen for the representation of the simulation outcomes as a reasonable and comprehensible way of identifying hot spots. These hot spots highlight deficits in the automated driving system functionality and outline requirements that must be refined to guarantee reliable system design. Heat maps could also serve as a potential way to facilitate safety assessment procedures in the future of ADAS and AD-function development. In this case, two maps were generated for the visualization of dropout time and minimum IoU.

Beginning with the findings for the minimum IoU, Figure 7 shows a nearly ideal flowing transition from high to low simulation outcomes for both rising obstacle heights and vehicle velocities. At the initial speed of 10 km/h, the overlapping area of predicted and real obstacle bounding box drops to minima between 88 and 84% for increasing road bump sizes. For the lowest bump height, the IoU consequently decreases, up to a velocity of 40 km/h and remains constant as of here. The same behavior accounts for the largest bump height, but at significantly lower IoU values and even reaching a minimum of 0% intersection. More interesting is the vehicle system response at medium obstacle size. While the minimum of 0% IoU is only reached at 50 km/h, the simulation result rises again to 22% at the highest tested velocity—a phenomenon which can only be observed at 0.1 m road bump height.

Proceeding to the results for the system dropout time, Figure 8 shows a clear performance degradation between the minimum and maximum ego-vehicle velocity with values dropping to half the reached maxima at the lowest speed of 10 km/h. Here, the time span without reliable distance estimation averages at around 2 s. By increasing the speed to 20 km/h, the dropout results remain similar for various obstacle sizes. However, when oscillating around the center of the valley, the dropout interval spans significantly shorter times for the smallest road bumps. At the center itself at a velocity of 40 km/h as well as for the highest tested speed, the time of unreliable distance estimations stabilizes again to around 0.9 and 1.35 s respectively.

Both generated maps can now be superimposed to unveil correlating issues. While the largest dropout time interval of around 2 s is found at the lowest tested vehicle velocity of 10 km/h, the minimum IoU consistently remains above 80% which might indicate an application deficit of the distance estimator. High sensitivity towards slight pitch angle changes can cause a massive drift in the estimation accuracy. At a rising vehicle speed of 20 to 40 km/h, the intersecting bounding box proportion decreases from 50 to 20%, but at a low disengagement interval of about 1 s. While the distance estimator only shows a slight lack in performance, the Kalman filter faces severe difficulties in predicting the correct target position. The most critical scenarios in this evaluation involve velocities over 40 km/h. At maximum obstacle size, the real and tracked bounding boxes’ IoU drops to zero. Furthermore, the dropout time is a critical parameter requiring further investigation. At the highest tested speed of 60 km/h, a disengagement time of 1.4 s can lead to a travelled distance of over 23 m without a plausible distance estimation. In this stage, the deficits of all parts within the processing pipeline show the greatest effects with issues in the tracking process as well as the estimator which must be tackled as a whole.

To further analyze the influencing factors leading to the phenomena observed in previous heat maps, it is worth analyzing the confounding vehicle dynamics factors. As stated in foregoing chapters, these comprise of the acceleration in *z*-direction and the pitch angular velocity. In addition, the *z*-velocity, pitch angle and pitch angular acceleration were examined. To ensure consistency in the evaluation, the results were again visualized in form of heat maps, illustrating the maximum values recorded during the test runs.

The results show the pitch angle as the greatest influencing factor on system performance. Especially the pitch angular velocity in Figure 9, which was also part of the validation criteria and assumed to be the main contributor to system failures, shows a pattern similar to the observed minimum IoU values, further supported by the acceleration in Figure 10. Both parameters respond with consistently low output values at 10 km/h. The analogies become even clearer at high speeds between 40 and 60 km/h where the acceleration shows a nearly identical pattern with a peak at 50 km/h which drops again at increasing velocity.

The dynamical parameters in *z*-direction additionally impact both minimum IoU and the dropout time interval and furthermore drastically account for passenger ride comfort. However, when investigating possible reasons for the presented heat patterns, the maximum *z*-acceleration alone is not sufficient to make comprehensive statements. Due to time constraints further investigation will be part of future research projects. Generic assumptions can still be made based on available results. The acceleration correlates with the Kalman filter performance at higher speeds and medium to high obstacle sizes between 0.1 and 0.12 m. The results in Figure 11 again demonstrate the effect of vertical impacts on the minimum overlap area between the real and estimated target position. Peak values are reached at 50 km/h and 0.1 m bump height as well as maximum velocity and obstacle size with maxima of 7.49 and 8.5 m/s${}^{2}$ respectively. The maximum recorded values nearly double at the highest driven velocity.

Finally, the last existing phenomenon of distance estimation performance decrease at 30 km/h can be explained by examining the vehicle pitch angle in Figure 12. While the angle only slightly rises between 10 and 20 km/h, a sudden surge from 0.06 to 0.074 rad can be seen when approaching the road obstacle at 30 km/h. Afterwards, at 40 km/h, the pitch angle flattens again to 0.068 rad at maximum obstacle height. It then further fluctuates around 50 km/h from 0.057 to 0.061 and back to 0.056 rad at medium bump size. In all analyses, the observation time was a massive impact factor which was not taken into consideration in this first discussion. Only the peak values extracted from each test run were researched while time continuous effects and the exact suspension setup are points of further investigation in upcoming projects.

## 4. Discussion

The results we obtained in this work can be summarized to three key findings:At a velocity of 10 km/h, the dropout time interval reaches its maximum at high IoU values of more than 80%, independent of the obstacle size. Issues in the distance estimation stage show the greatest negative impact.At velocities between 20 and 40 km/h, the dropout time is minimal. The IoU is highly dependent on the bump height. Deficits in the Kalman filter application influence the results to the major part.At velocities above 40 km/h, the dropout time reaches an average of about 1.4 s. Dependent on the object size, the IoU drops to zero. The results are sensitive to both the distance estimator and Kalman filter adaption process.

In addition to the analysis of the results and the influencing algorithms, we are now suggesting possible solutions and approaches to increase system reliability in the future. All suggestions affect research areas which are already investigated, but mostly with a focus on flat-world models. The first solution is fusing the monocular camera with other sensor technologies to generate reference data. This approach creates a more reliable system architecture with redundant signal paths to filter outliers within the sensor stream. However, thorough plausibility analysis is required to ensure streamlining the correct sensor output quantities and setting robust threshold values when comparing the parameters of multiple sensor systems. In the relevant ISO standards for functional safety (ISO 26262) and safety of the intended functionality (ISO/PAS 21448), redundancy is mandatory and must be guaranteed to fulfill the system safety requirements.

Secondly, installing active suspension systems can support compensating the dynamical impact of road anomalies, where the suspension can adapt to variations in the road surface structure to mitigate failures in the vehicle’s perception process. However, to adapt the ride height to the upcoming anomaly, the system must detect the road obstacle in advance. The exact time of impact as well as the proportions of the irregularity must be known for the controller to compute the correct damping pressure. However, if the detection fails, the vehicle must deal with an uncompensated impact, whose consequences have been analyzed in this study.

The results that we have seen all result from disturbances in the acquired image data and the time-discrete variance in the continuous data stream. By excluding the dynamical influences of the driving environment, a more stable vision process can be achieved. The third solution we propose is the adaption of image stabilizing algorithms. Here, usually one object in the scene is used as a reference and maintained in the same position by adjusting the forwarded image frame. The presented image only shows a fraction of the actual sensor input and is varied according to the scene dynamics. While this technique allows for the detection and tracking of objects in stable image data, the reaction time might be reduced as the algorithm tries to maintain consistent output. In a highly dynamical environment with rapid vehicle state changes, such as in evasive maneuvers or the presence of road irregularities, the solution might introduce lag to the processing pipeline mitigating appropriate reaction times or even hiding relevant parts of the actual recorded frame. Furthermore, we also conducted a study on the sensitivity of image stabilizers and the effect on the reliability and performance of the YOLOv3 and Mask RCNN object detection networks and detected several corner test cases. While the overall image is steadier with a stabilizer, especially on road bumps the output picture is blurred when facing *z*-acceleration values of more than 3 m/s${}^{2}$. None of the networks tested could detect objects in the blurred data, which is a phenomenon that must be further investigated. The interested reader is referred to contacting the authors in case more information on the additional study is required.

A major issue in the context of dynamical scenarios is the pitch angular velocity. The distance estimation algorithm is highly sensitive to rapid changes in the vertical axis as the system is calibrated to the initial camera mounting angle at a standstill, so zero vehicular pitch. Instead of a steady angle application, the car’s pitch could be compensated in real time to account for a more robust system architecture. For the realization, on the one hand high performance IMUs could be integrated to directly feed the estimation processor and compensate state changes in real time. On the other hand, further supporting a priori processes such as including a horizon or lane marking tracker for solid ground-plane estimation might aid designing solid algorithms. However, as Figure 13 extracted from our collected data set shows, the real world offers a variety of challenging scenarios. In this example, neither a clear horizon line nor lane markings can be found in the picture. Furthermore, the road bump obstacle is occluded by a parked vehicle. Corner cases such as these must be considered during the system development phase and tested to ensure reliability in any environment when aiming at level 4 and 5 on the SAE automation scale.

A final possible solution we elaborated could be the use of model predictive controllers (MPCs) computing the possible vehicle response to upcoming and detected road anomalies to digitally compensate the sensor system response. However, as we highlighted before, the obstacle must be detected in advance. In addition, MPCs are cost-intensive and require a vast amount of computing power during execution.

## 5. Conclusions

In the development process of automated vehicles, the reliability of sensor and processing systems is a key factor for the realization of safe self-driving. Not only designing the systems is a challenge, but also proving system safety and reliability is essential for the release of highly safety-critical transport systems which are operating on public roads. In this regard and to the best of our knowledge, we were the first ones to investigate and to prove that there is a clear connection and correlation between vehicle (vertical) dynamics and the performance of vision systems. We therefore demonstrated that the discipline of automotive engineering is still eminently relevant from the concept to the component development phase and from processor to consumer acceptance testing. Computer science and engineering must come together in the interdisciplinary field of automated driving to maximize efficiency, especially in the perceptual process of a car generating its own version of the surrounding world, interacting with human beings.

While latest research projects mostly focus on the development of AI-based vehicle vision systems in flat-world models, we highlighted the importance of including scenarios provoking corner cases for vehicle dynamics to estimate the performance boundaries of vehicle detection, tracking, position prediction and distance estimation algorithms. Even though research was already conducted in the area of road anomaly detection and compensation, critical scenarios can arise in the event of undetected road bumps, especially at higher speeds and larger object sizes or scenarios that were not considered to fall into the car’s operational design domain.

Finally, we discovered IPG CarMaker to be a suitable development tool for the application of vision systems in automated driving systems. Hyper-realistic graphics are not necessarily required for the investigation of vehicle dynamics influence on the perception process. We suggest to divide the design procedure into two separate stages: (1) It is recommended to design and train the object detector in a high fidelity graphics environment to ensure the algorithm learns an object representation as close to the real world as possible. (2) With our experiments we showed that a neural network fully trained on real-world images can then be transferred to the CM simulation without a decrease in performance. Therefore, we suggest to then take the trained algorithm from step (1) and apply it to the virtual environment without major adjustments to the network itself. CM then provides an efficient solution for the development of e.g., active suspension architectures or real-time controllers relying on the detection algorithm’s output information. Consequently, we present a cost-effective and economically efficient way of combining already existing development environments.

On the hardware side, monocular cameras offer a promising solution for the automotive industry and eventually product consumers to reduce cost and can serve as a redundant reference path in safety-critical vehicle systems. However, they carry a risk that must be considered and investigated. Due to lacking triangulation capabilities compared to a stereo camera, especially distances must be digitally computed without relying on intrinsic parameters and sensor positioning. For future analyses we are planning to extend the scope of our work by making use of stereo camera devices as well as fusing camera vision with LiDAR sensors to investigate the impact of vehicle dynamics on the validity of diverse sensor input data.

## Figures and Tables

**Figure 1 sensors-19-04870-f001:**
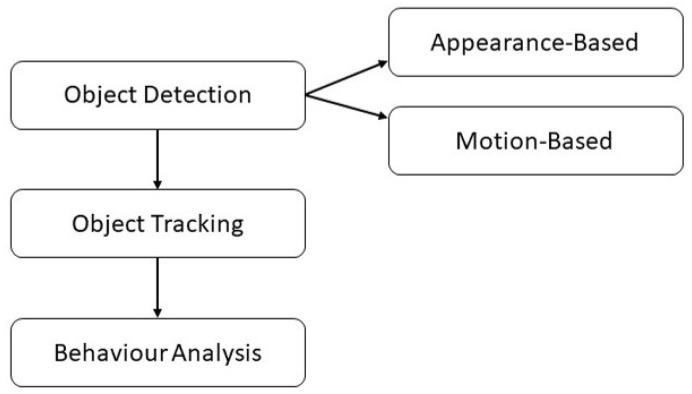
Three stages of camera-based vehicle perception.

**Figure 2 sensors-19-04870-f002:**
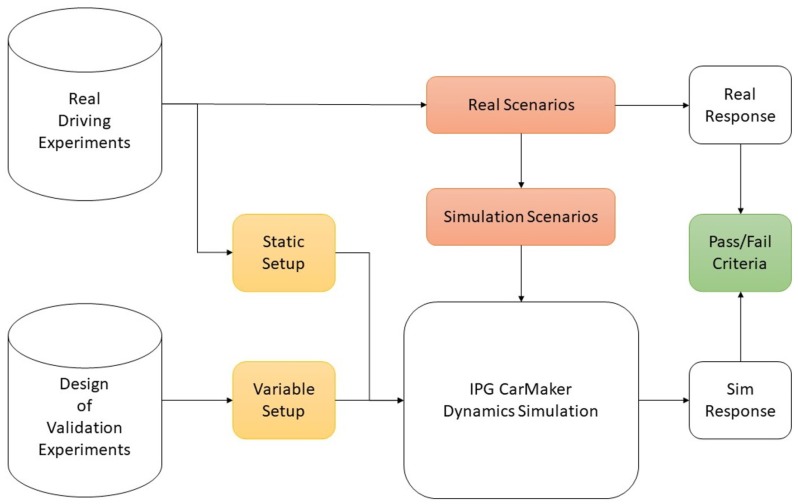
IPG CarMaker vehicle dynamics simulation model validation procedure.

**Figure 3 sensors-19-04870-f003:**
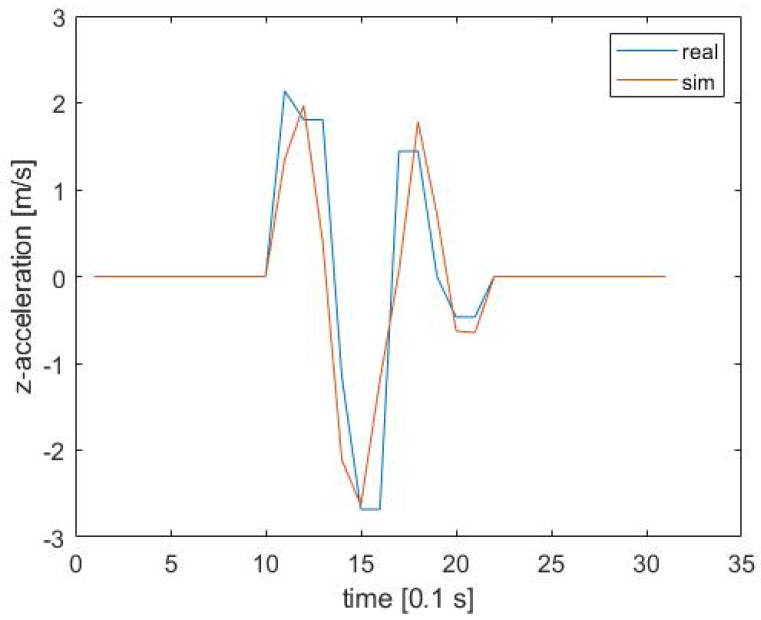
Comparison of vertical acceleration values recorded during the real-world field trial and the one obtained using simulation in Scenario 1 ($R^{2}$ = 69.76%, RMSE = 0.52 m/s${}^{2}$).

**Figure 4 sensors-19-04870-f004:**
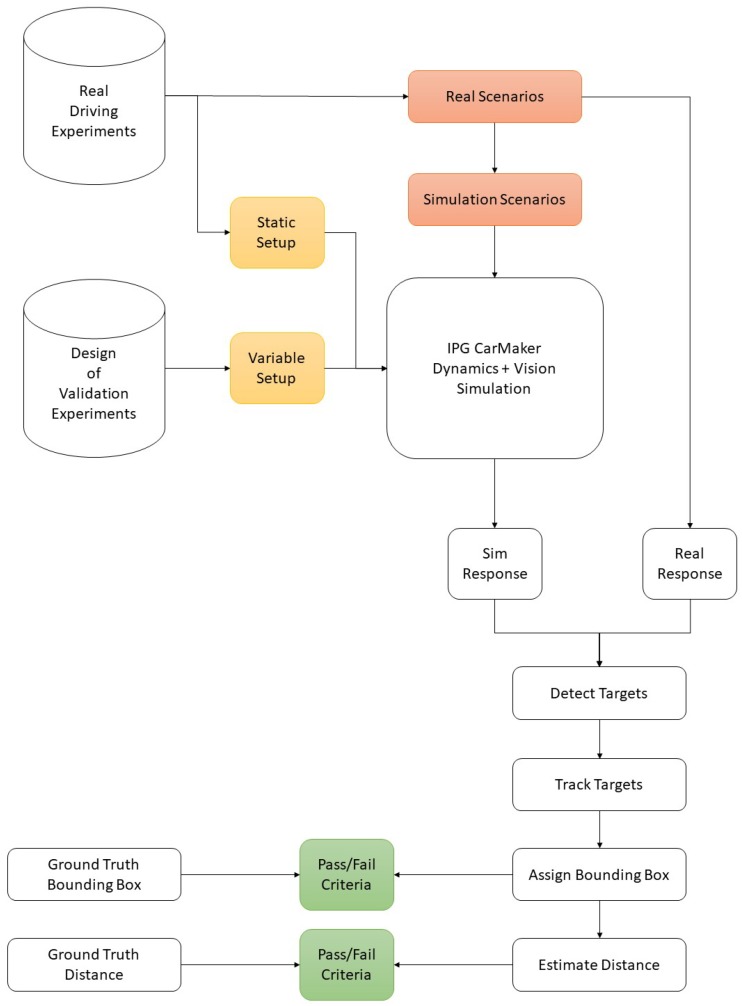
IPG CarMaker vehicle vision simulation model validation procedure.

**Figure 5 sensors-19-04870-f005:**
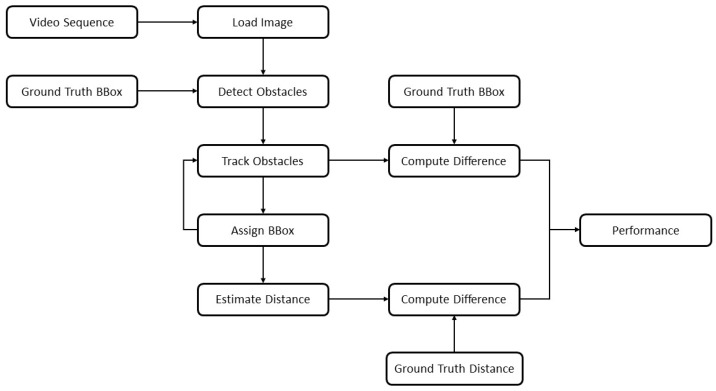
System environmental perception evaluation framework.

**Figure 6 sensors-19-04870-f006:**
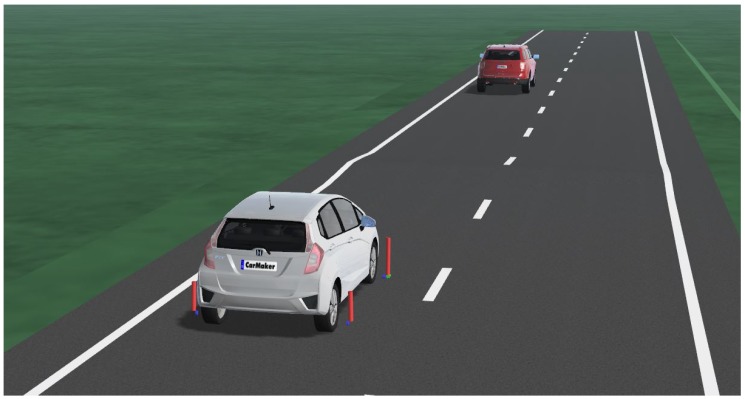
Performance evaluation scenario including ego, target vehicle and a road bump obstacle.

**Figure 7 sensors-19-04870-f007:**
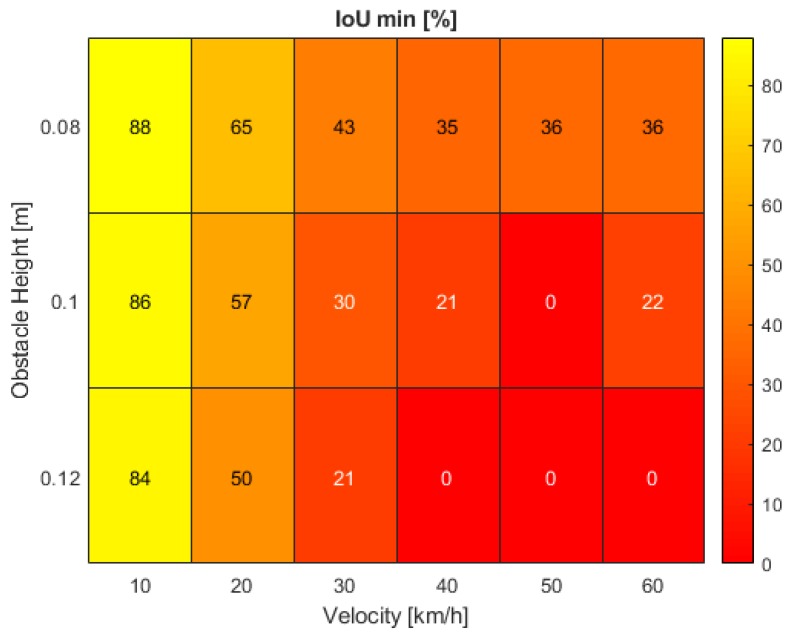
Minimum intersection over union (IoU) of the real and tracked vehicle positions at various bump obstacle heights and vehicle velocities in the simulation.

**Figure 8 sensors-19-04870-f008:**
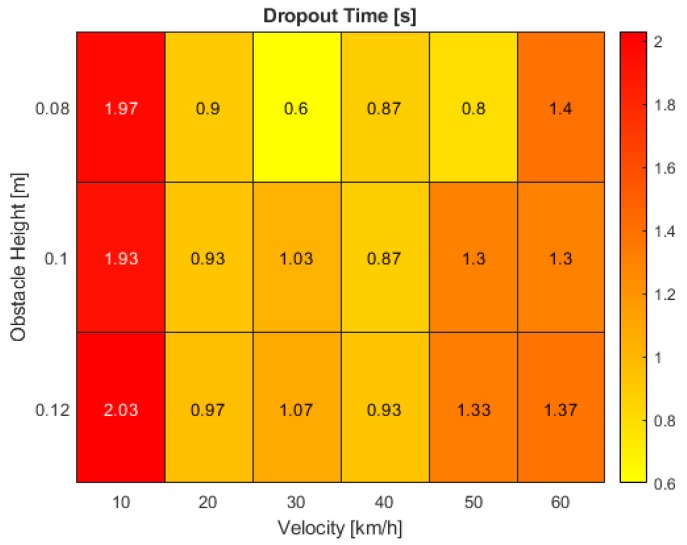
Distance estimation dropout time at various bump obstacle heights and vehicle velocities in the simulation.

**Figure 9 sensors-19-04870-f009:**
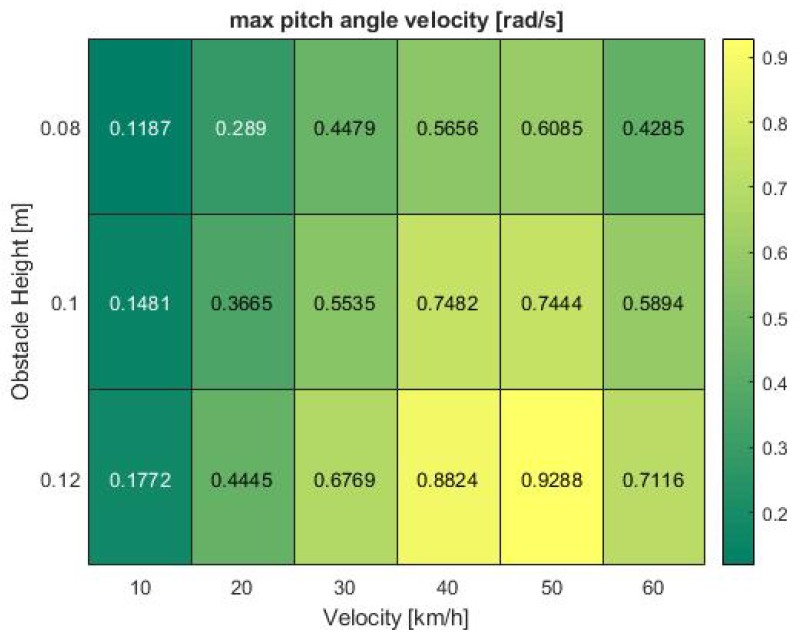
Pitch angle velocity at various bump obstacle heights and vehicle velocities in the simulation.

**Figure 10 sensors-19-04870-f010:**
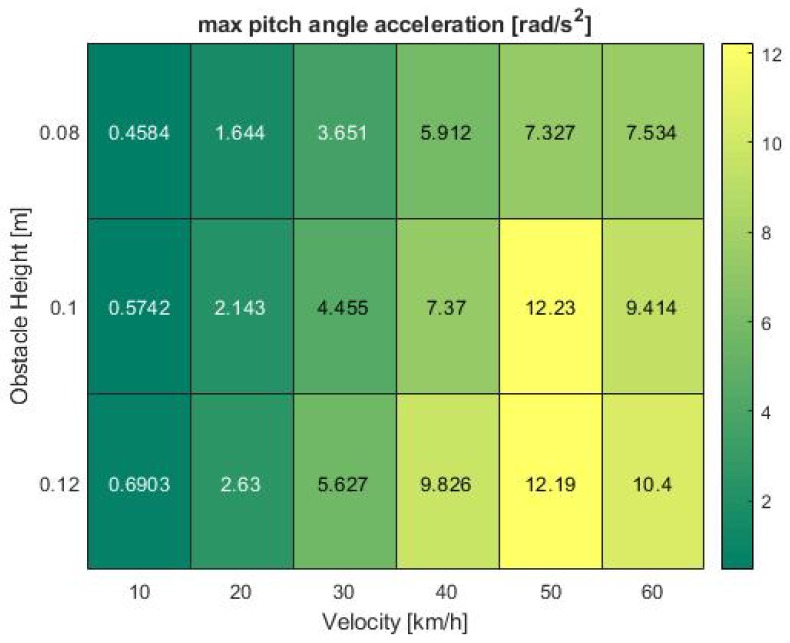
Pitch angle acceleration at various bump obstacle heights and vehicle velocities in the simulation.

**Figure 11 sensors-19-04870-f011:**
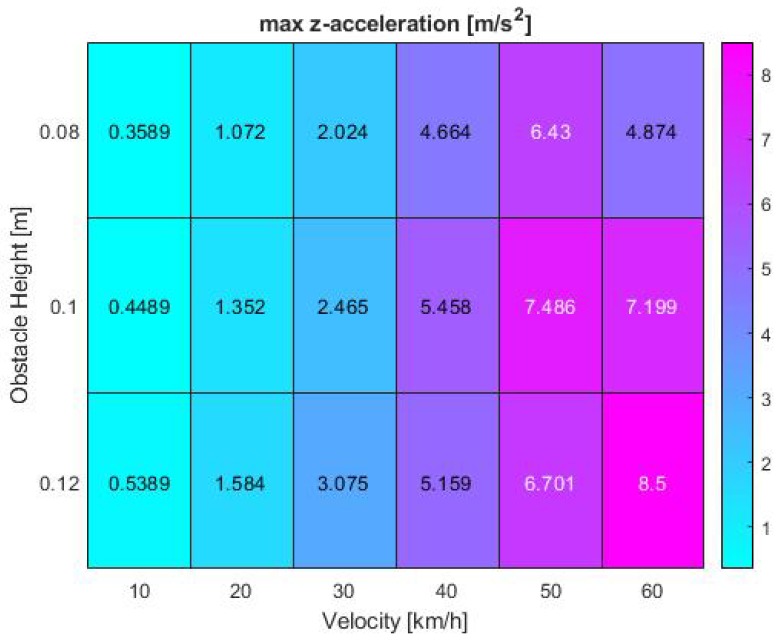
Vehicle *z*-acceleration at various bump obstacle heights and vehicle velocities in the simulation.

**Figure 12 sensors-19-04870-f012:**
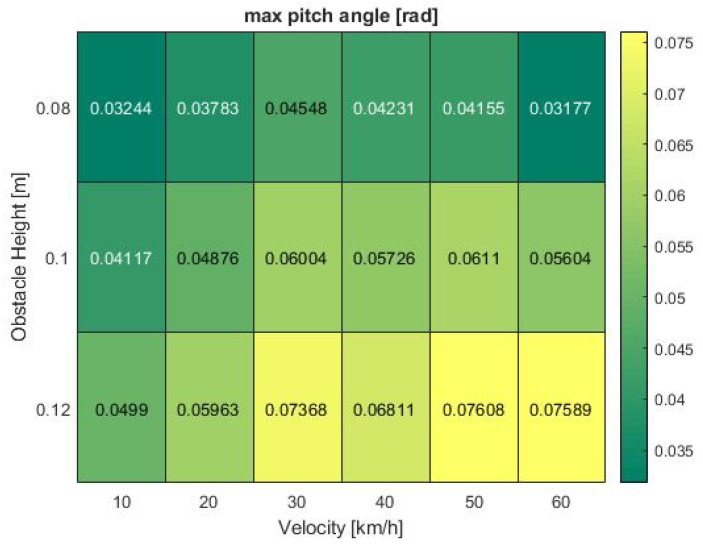
Pitch angle at various bump obstacle heights and vehicle velocities in the simulation.

**Figure 13 sensors-19-04870-f013:**
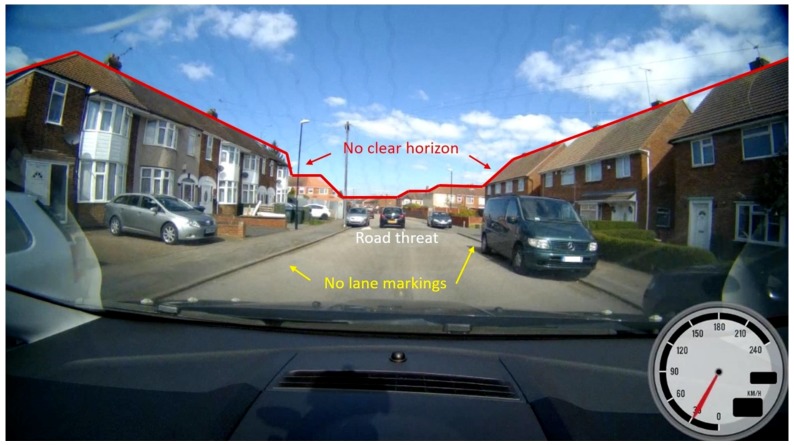
Corner case driving scenario including unclear horizon line, non-existing lane markings and obstructed road bump obstacle.

**Table 1 sensors-19-04870-t001:** Research vehicle and tire parameters.

$l_{f}$	1.2 m
$l_{r}$	1.45 m
$I_{\psi }$	2325 kgm${}^{2}$
*m*	1580 kg
$r_{e}$	0.32 m
$I_{\omega }$	2 kgm${}^{2}$
$t_{\omega f}$	1.5 m
$t_{\omega r}$	1.5 m

**Table 2 sensors-19-04870-t002:** Parameterization of 5 road bump scenarios for the vehicle dynamics model validation.

Scenario	$a_{z,\min}$/$a_{z,\max}$ [m/s^2^]	$\omega_{y,\min}$/$\omega_{y,\max}$ [s^−1^]	$v_{\mathrm{abs},\min}$/$v_{\mathrm{abs},\max}$ [km/h]
1	−2.5/2	−0.3/0.4	21/26
2	−2.5/2	−0.35/0.35	19/24
3	−3/2	−0.5/0.3	25/29
4	−3/2.5	−0.45/0.5	27/31
5	−3.5/3	−0.35/0.5	26.5/31

**Table 3 sensors-19-04870-t003:** Minimum validation criteria and experimental results.

Parameter	$R_{\min}^{2}$/$R^{2}$	${\mathrm{RMSE}}_{\max}$/RMSE	Result
$a_{z}$	67/73.5%	1/0.51 m/s^2^	Pass
$\omega_{y}$	67/67.18%	0.1/0.1 s^−1^	Pass
$v_{abs}$	67/73.15%	1/0.92 m/s	Pass

**Table 4 sensors-19-04870-t004:** Monocular camera sensor model parameters.

$d_{x,y,z}$	2.7, 0, 1.24 [mm]
$r_{x,y,z}$	0, 4, 0 [deg]
FoV	148 [deg]

**Table 5 sensors-19-04870-t005:** Key velocities of 5 road bump scenarios for the vehicle model validation.

Scenario	$v_{\min}$/$v_{\max}$ [km/h]	$v_{\mathrm{bump}}$ [km/h]	Driving Profile
1	27/30.5	28.5	Constant
2	12/22	20	Acceleration
3	31/33	31.5	Constant
4	27/34	28	Deceleration
5	10/28	-	Diverse

**Table 6 sensors-19-04870-t006:** Precision and recall of four different networks on simulation data.

Network	Precision	Recall
YOLOv3	92.11%	93.33%
Mask RCNN	100%	97.67%

**Table 7 sensors-19-04870-t007:** Minimum validation criteria and experimental results.

Parameter	${\mathrm{Deviation}}_{\min}$/${\mathrm{Deviation}}_{\mathrm{res}}$	Result
$BBox_{height,diff,avg}$	90/93.98%	Pass
$BBox_{width,diff,avg}$	90/96.16%	Pass
$d_{est,diff}$	90/95.34%	Pass

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
