# Peer review of "The Correlation between Vehicle Vertical Dynamics and Deep Learning-Based Visual Target State Estimation: A Sensitivity Study"

_sensors, 2019, doi:10.3390/s19224870_

Round 1

Reviewer 1 Report

The papers is a very lengthy but comprehensive study on the correlation between vertical dynamics and DL visual target state estimation.

The introduction can be split been actual introduction and state-of-the-art.

There are references to related works, but maybe too many references.

The methodology and results sections are very well written with very relevant  information useful to fully understand the case in study.

Conclusions and proposals are nonetheless so novel and could have been reached empirically 

Author Response

Thank you very much for taking your time and providing valuable feedback for the research we conducted. Your appreciation for the time we spent on investigating the correlation between vehicle vertical dynamics and visual target state estimation using deep learning algorithms means a lot to us. Following your recommendation, we split the introduction into ‘Introduction’ and ‘State-of-the-Art’ making it easier to comprehend and follow our elaboration of contributors to the field and state of the art methods used. Taking the recommendations of your fellow co-reviewers into consideration, we decided to stick to the existing list of references in the first place and narrow it down to essential literature in upcoming publications.We would also like to express our gratitude for your kind words about the methodology and results sections of our work that we put a lot of time and effort into. In future projects of ours the conclusions we draw will be extended and undermined with findings from studies including a second camera for stereo vision analysis, a more in-depth driving dynamics analysis as well as further real-world experiments for verification and validation of the results. 

Thank you very much for sharing your thoughts with us. This piece of work means a lot to us and we would be honored to involve your valuable opinion in future papers we produce.

Reviewer 2 Report

This very interesting article talks about the influence of road anomalies and vehicle suspension on the performance of detecting and tracking driving objects.

It is very well structured and very educational. The concepts are well argued and the solutions proposed.

The bibliography is extensive and on the whole quite recent.

The main components analyzed are at the intersection of artificial vision and AI, more particularly deep learning.

The introduction is well presented and offers a good state of the art of the discipline with very good references.

A lot of work is currently being done on artificial vision and deep learning links. The originality of this work consists in modeling behaviors in front of anomalies. Most of the work is obtained by simulation, I would have liked to see more detailed problems of implementation, performance, even the discussion on the use of stereo cameras may not be enough and it seems to me personally that it would be a shame not to take advantage of the advantages it offers...

I think this work will be of interest to researchers and engineers in the field and its publication is very relevant

Author Response

Thank you very much for taking your time and providing valuable feedback for the research we conducted. Your appreciation for the time we spent on investigating the correlation between vehicle vertical dynamics and visual target state estimation using deep learning algorithms means a lot to us.

We would also like to express our gratitude for your kind words about the general concept and solutions of our work that we put a lot of time and effort into. Undermining the core ideas with relevant examples from the literature was very important to us, especially with regards to deep learning techniques which are part of a lively discussion in the self-driving research area. We would like to thank you as well for your appreciation towards this part of the investigation.

We are glad to contribute to state of the art knowledge in such a competitive field as automated driving and to initiate a discussion by linking computer vision with road anomalies. We completely agree with you on the potential and advantage an extension of the study could have. Unfortunately, it was not possible for us to cover more aspects - like e.g. the inclusion of stereo cameras - and extend the scope of our work. However, the extension of real-world and simulation experiments with different sensor technologies will be part of our future work. We are also highly interested in the sensor fusion of LiDAR and different camera devices to analyse the impact it might have on the validity of diverse input data. We decided to focus on monocular camera in our first approach as the devices are widely used within the industry and pose a cost-effective solution for automated driving functions. To clearly emphasise our attention to the reader, we extended the very last sentence of our work as follows: 

‘For future analyses we are planning to extend the scope of our work by making use of stereo camera devices as well as fusing camera vision with LiDAR sensors to investigate the impact of vehicle dynamics on the validity of diverse sensor input data’.

Thank you very much for sharing your thoughts with us. This piece of work means a lot to us and we would be honoured to involve your valuable opinion in future papers we produce.

Reviewer 3 Report

Comments:

This paper  analyses and discusses the correlation between vehicle vertical dynamics  and Deep Learning-based visual target state estimation. My comments are as follows.

In Section 1, the contribution of this manuscript should be highlighted to clearly analyze advantages. Many previous researches have focused on the correlation between vehicle and visual target. More state-of-the-art literature should be added and discussed. Many previous researches have focused on the correlation between vehicle and visual target. What are the advantages of the proposed method?  In Section “Experimental Evaluation”, more analysis for the proposed method is encouraged. Meanwhile, some experimental comparisons are encouraged. And the experiment may be analyzed comparatively to show its significance.

Author Response

Thank you very much for taking your time and providing valuable feedback for the research we conducted.

Following your recommendation, we clearly outlined our approach and contribution to this highly debated field of research by adding the following paragraph to the end of our introduction chapter:

‘To the best of our knowledge this is the first time a study on the impact of vehicle vertical dynamics on the performance of vehicle detection, tracking and distance estimation is performed. Furthermore, it is the first contribution applying a perception pipeline based on state-of-the-art deep learning-based object detectors to a fully validated IPG CarMaker simulation environment to accelerate the verification and validation of automated driving functions in the system development process’.

Further analysing of state-of-the art literature will be part of future studies we are conducting. Here, we are planning to also take other sensor technologies into consideration, like stereo camera devices or the analysis of a sensor fusion with LiDAR or Radar. In addition, we also want to extend real-world and simulation studies in the next chapter of our research. Unfortunately, it was not possible for us to cover these aspects in our current work.

We highly appreciate your comments regarding experimental comparison and added the quantity of road anomalies we were facing in real-world experiments as well as the amount of road anomalies we analysed. Therefore we added following note to the real-world driving data section: 

‘Furthermore, the set contains around more than 100 road bump objects with correlating tri-axial acceleration data and a multiple of the quantity covering anomalies with inertial vehicle measurements’. 

We agree that more information demonstrates significance of the work itself. To further highlight the advantages of our proposed methods, we added details to the introduction chapter of this paper:

‘It is the first contribution applying a perception pipeline based on state-of-the-art deep learning-based object detectors to a fully validated IPG CarMaker simulation environment to accelerate the verification and validation of automated driving functions in the system development process. The rest of the paper structures as follows: After describing methods and tools used in our research, we present the experimental results from several real-world driving trials. Then, the computational results of our study are analysed, followed by conclusions where the advantages of our proposed methodology are highlighted and finally the scope of upcoming projects and future research directions’ 

This clearly guides the reader to the advantages of our approach, worked out in the conclusion part. We also included a statement on the work we have planned to deliver a more in depth follow-up investigation to deliver more information for the actual automated driving development process. Please find this paragraph added to the very last sentence of our work:

‘For future analyses we are planning to extend the scope of our work by making use of stereo camera devices as well as fusing camera vision with LiDAR sensors to investigate the impact of vehicle dynamics on the validity of diverse sensor input data’.

Thank you very much for sharing your thoughts with us. This piece of work means a lot to us and we would be honoured to involve your valuable opinion in future papers we produce.